# Position: Rethinking Explainable Machine Learning as Applied Statistics

Sebastian Bordt [1]    Eric Raidl [2]    Ulrike von Luxburg [1]

## Abstract

In the rapidly growing literature on explanation algorithms, it often remains unclear what precisely these algorithms are for and how they should be used. In this position paper, we argue for a novel and pragmatic perspective: Explainable machine learning needs to recognize its parallels with applied statistics. Concretely, explanations are statistics of high-dimensional functions, and we should think about them analogously to traditional statistical quantities. Among others, this implies that we must think carefully about the matter of interpretation, or how the explanations relate to intuitive questions that humans have about the world. The fact that this is scarcely being discussed in research papers is one of the main drawbacks of the current literature. Moving forward, the analogy between explainable machine learning and applied statistics suggests fruitful ways for how research practices can be improved.

## 1. Introduction

Despite a growing literature on explanation algorithms, their use cases, and evaluation, the field of explainable machine learning remains in a pre-paradigmatic state. This is because there is no agreement on the meaning of the basic terminology in the field — what is an explanation? when is a model interpretable? — as has been repeatedly noted in different papers (Doshi-Velez & Kim, 2017; Lipton, 2018; Murdoch et al., 2019; Freiesleben & König, 2023).

In this position paper, we argue for a novel perspective on explainable machine learning: The field needs to recognize its parallels with applied statistics. By applied statistics, we mean the use of statistical theory, methods, and tools to analyze data and solve real-world problems (Wilcox & Rousselet, 2023; Angrist & Pischke, 2009; Cox & Donnelly,

2011; Cox, 2018; Franconeri et al., 2021). Both explainable machine learning and applied statistics aim to summarize the behaviour of high-dimensional objects — functions in explainability, and datasets and their respective probability distributions in statistics (Wasserman, 2004). Moreover, both fields share a common goal: determining how relevant questions about the real world can be answered using such summaries.

The advantage of our perspective is that applied statistics is a fairly established field of research, which means that research practices in explainable machine learning can be improved by drawing appropriate analogies. In this paper, we argue that it is especially important to distinguish between the following two concepts:

(a) The mathematical form and properties of an explanation algorithm.

(b) The matter of its interpretation.

Mathematically, explanation algorithms have the form of a functional — a function of functions (Rudin, 1991). They take a function and map it to a simpler object, often a vector. One might say that the functional "explains" a function by reducing its complexity. However, not all dimension-reducing functionals have explanatory value. For this to be the case, the functional also needs to formalize an intuitive concept, a nuanced matter debated in the philosophy of science (Justus, 2012). If this is the case, then we say that the functional *has an interpretation*. Traditional statistical quantities like the $p$-value, confidence intervals and visualizations usually have an interpretation.

In the current literature on explainable machine learning, papers often introduce novel explanation algorithms without discussing their interpretation. This is problematic in itself because "explanations" without an interpretation miss the point. It is additionally problematic since historically, even relatively simple statistics that *have* an interpretation have frequently been misinterpreted, especially by applied scientists and decision-makers (Wasserstein & Lazar, 2016). Consequently, it is not surprising that the outputs of explanation algorithms have frequently been misinterpreted as well (Molnar et al., 2020).

[1]University of Tübingen, Tübingen AI Center, Germany [2]University of Tübingen, Germany. Correspondence to: Sebastian Bordt <sebastian.bordt@uni-tuebingen.de>.

*Proceedings of the $42^{nd}$ International Conference on Machine Learning*, Vancouver, Canada. PMLR 267, 2025. Copyright 2025 by the author(s).

The analogy between explainable machine learning and applied statistics gives rise to several concrete recommendations for improving research practices. For one, explanation algorithms should be designed to answer specific questions, as in Schut et al. (2023) or Arditi et al. (2024). Moreover, while it is a worthy goal to design explanations that are useful to end users (Liao & Varshney, 2021), working with statistics requires a certain degree of expertise, and we must recognise that this is also the case for many of the techniques proposed in explainable machine learning. In addition, our perspective has implications for the relationship between explanations and other statistics of machine learning models, such as fairness and robustness measures, and for the role of benchmark datasets (Section 6).

Until now, the literature on explainable machine learning has mainly tried to connect explanation algorithms to insights from the social sciences and psychology, where an explanation is mainly conceived of as an answer to a why-question (Mittelstadt et al., 2019; Miller, 2019). In this paper we take a broader, but also more concrete perspective, looking at statistical answers to intuitive questions.

To be concise, this paper focuses on post-hoc explanation algorithms, which comprise the bulk of the research in explainable machine learning over the last decade. However, the analogy between explainable machine learning and applied statistics applies broadly. In Section 7, we briefly discuss how it applies to mechanistic interpretability.

**This paper takes the position that explainable machine learning should be considered as applied statistics for high-dimensional functions. In particular, post-hoc explanations are fundamentally equivalent to traditional statistical approaches. Recognizing this resolves many of the foundational debates within the field. It also illuminates why certain approaches in explainable machine learning have remained largely unsuccessful and suggests how research practices can be improved.**

## 2. Post-Hoc Explanation Algorithms are Statistics of Functions

This section highlights the similarities between post-hoc explanation algorithms and traditional statistical quantities.

**Notation.** $\mathcal{D}$ denotes a probability distribution on $\mathbb{R}^d$ and $f : \mathbb{R}^d \to \mathbb{R}$ a function. We let $\mathcal{P}(\mathbb{R}^d)$ denote the space of all probability distributions on $\mathbb{R}^d$ with Borel $\sigma$-algebra and $\mathcal{F}(\mathbb{R}^d)$ the space of all functions $f : \mathbb{R}^d \to \mathbb{R}$.

### 2.1. Background: Statistics of probability distributions and datasets

We begin by introducing some notation from the statistics literature.[1]

**Definition 2.1** (Statistic of a Probability Distribution)**.** A statistic of a probability distribution is a functional $F : \mathcal{P}(\mathbb{R}^d) \to \mathbb{R}^k$.

A statistic of a probability distribution takes a complex object — a multi-dimensional probability distribution — and maps it to a simpler object — a single number or a vector. Examples of statistics of probability distributions are (Wasserman, 2004):

1. The *mean*, *median*, and *moments* of the distribution.

2. The *minimum and maximum values* that can occur under the distribution.

3. The *coefficients* $\phi$ of the linear model that best approximates the distribution.

4. Any other *population parameter* (Fu & Li, 1993).

Closely related to statistics of probability distributions are the statistics of high-dimensional datasets.

**Definition 2.2** (Statistic of a Dataset)**.** A statistic of a dataset is a function $F : \mathbb{R}^{d \times m} \to \mathbb{R}^k$.

The mathematical definition is slightly different, but the principle is the same: A statistic of a dataset takes a complex object — a dataset of $m$ data points in $\mathbb{R}^d$ — and maps it to a simpler object — a single number or a vector. Examples include the $p$-value, the F-statistic, visualizations, and statistical estimators (Wasserman, 2004, Chapter 6). In the statistics literature, these statistics are also known as sample statistics (Cox, 2006, Chapter 2).

### 2.2. Explanation Algorithms: Statistics of functions

In analogy to the concept of a statistic of a probability distribution, we now introduce the notion of a statistic of a function.

**Definition 2.3** (Statistic of a Function)**.** A statistic of a function is a functional $F : \mathcal{F}(\mathbb{R}^d) \times \mathcal{P}(\mathbb{R}^d) \times \mathbb{R}^d \to \mathbb{R}^k$.

A statistic of a function takes a complex object — a high-dimensional function $f$ — and maps it to a vector. We allow the statistic to additionally depend on a probability distribution and a specific data point $x$. Examples of statistics of functions are

---

[1] For the usage of the term "statistic" in the statistics literature, see, for example, Cox (2006, Chapter 2) and Schervish & DeGroot (2012, p. 381). In the machine learning literature, statistics of distributions are also referred to as properties (Steinwart et al., 2014; Frongillo & Kash, 2015).

1. The *generalization error* of the function $f$ with respect to a distribution $\mathcal{D}$ and loss function $l(x, y)$.

2. *Fairness measures* (Barocas et al., 2023).

3. The *approximation error* of $f$ with respect to some simpler function class $\mathcal{I}$ (Dziugaite et al., 2020).

4. The *derivative* of the function $f$ (Szegedy et al., 2014).

5. *SHAP* values (Lundberg & Lee, 2017):

$$F_i(f, \mathcal{D}, x) = \sum_{S \subseteq [d] \setminus \{i\}} \frac{|S|!(d-|S|-1)!}{|d|!} [\mathbb{E}_{\mathcal{D}}(f|x_{S \cup \{i\}}) - \mathbb{E}_{\mathcal{D}}(f|x_S)].$$

6. *LIME* (Ribeiro et al., 2016; Agarwal et al., 2021).

7. *Layer-wise relevance propagation*[2] (Bach et al., 2015).

8. *Grad-CAM* and its variants (Selvaraju et al., 2017; Chattopadhay et al., 2018).

9. The closest point to $x$ with a different label than $x$:

$$\underset{z \in \{y|f(y) \neq f(x)\}}{\arg \min} ||x - z||_1.$$

10. *Perturbation-based* explanations (Covert et al., 2021).

In summary, Definition 2.3 captures most, if not all, local post-hoc explanation algorithms. It also captures various other properties of functions, such as fairness- and robustness metrics. Note that when we talk about an *algorithm*, we mean the function implemented through that algorithm.

The fact that explanations are functionals of the learned predictor has previously been noted in the literature (Fisher et al., 2019; Marx et al., 2023; Scholbeck et al., 2024). In this work, we build on this insight to discuss the relationship between the mathematical aspects of explanation algorithms and their interpretation.

## 3. The Analogy Between Explainable Machine Learning and Applied Statistics

We have seen that post-hoc explanations and other properties of classifiers can be seen as statistics of functions and that there is a formal similarity between statistics of functions and statistics of probability distributions and datasets. This suggests that we should think about explanations and traditional statistical quantities along analogous lines and that explainable machine learning should be considered statistics for high-dimensional functions.

---

[2] Explanation algorithms for deep neural networks often depend on the parametrization of the function. For them, $\mathcal{F}(\mathbb{R}^d)$ in Definition 2.3 would need to be replaced with the parameter space of the deep neural network.

Historically, researchers in statistics have studied how the properties of probability distributions and large datasets can be summarized using distribution and dataset statistics (Stigler, 2002). In explainable machine learning, researchers have studied how the properties of learned functions can be summarized using explanations. In both cases, the principle is the same: We take a high-dimensional object, map it to a low-dimensional space, and then use the resulting statistical value(s) to answer relevant questions that we are interested in.

As we argue in the following Section 4, the analogy between explainable machine learning and statistics extends towards the crucial question of how to interpret explanations in applications. Before this, we want to briefly discuss two important insights.

**Insight 1:** Post-hoc explanation algorithms are statistics of functions, regardless of whether they are useful to an end user, fulfill a purpose, or admit an interpretation.

One of the foundational questions in explainable machine learning is the meaning of the term "explanation". When should we say that something is an explanation? The formal framework helps to take a step back and see that explanations are, first of all, statistics of functions. Whether the statistic is faithful, useful to an end user, or helpful at any other task is something that needs to be established in addition.

**Insight 2:** A large part of the literature in explainable machine learning studies the mathematical and computational properties of statistics of functions.

The mathematical framework introduced in Section 2 can help us to understand and categorize research questions in explainable machine learning. In particular, we can ask whether a research question can be fully expressed within the mathematical framework. In other words: Which research in explainable machine learning studies the mathematical properties of high-dimensional functionals? It turns out that a large part of the literature studies such questions. For example, many papers study the sensitivity of a statistic $F$ with respect to $f$ and $x$ (Slack et al., 2020; Anders et al., 2020; Lakkaraju et al., 2020), or the behaviour of a statistic $F$ on a subset of functions $\mathcal{I} \subset \mathcal{F}(\mathbb{R}^d)$ (Garreau & von Luxburg, 2020; Bordt & von Luxburg, 2023). Similarly, many papers study the computational aspects of statistics of functions, for example, of Shapley values (Jethani et al., 2021; Covert et al., 2023; Wang et al., 2025).

Research on the mathematical properties of statistics can be useful. In particular, research in statistics also has strong mathematical aspects (Larsen & Marx, 2005). However, it is also important to highlight what this research usually does not discuss. Namely, what are the real-world questions that can be answered with the explanations, and why?

# 4. Statistics and their Interpretation

In this section, we ask how the mathematical aspects of explanations are related to the real-world questions that explanations are designed to answer. This is one of the most important questions in explainable machine learning. Because it is scarcely addressed in research papers, we want to take some time to discuss its philosophical underpinnings.

Both in applied statistics and explainable machine learning, the starting points are intuitive questions humans have about the real world or the model (Liao et al., 2020). For example,

1. How does this model *make* predictions?

2. Should I *trust* the predictions of the model?

3. Which input features are *important* to the prediction?

4. Would model performance *improve* if we stack more layers?

5. Is the model *relying on spurious artifacts* in the input image?

6. How do these genetic variants *influence* disease risk?

7. What is the *impact* of carbon pricing on economic growth?

These questions are often broad and exist outside any formal framework. Following works in the philosophy of science, we say that they express *intuitive concepts* (Justus, 2012). Research often starts with intuitive concepts and questions and attempts to formalize them.

Here, we face a reverse situation: How do statistics of functions — formal mathematical objects — relate to intuitive concepts, such as whether one should trust a model's predictions? This is discussed in the following Section 4.1. We also introduce the following terminology.

**Definition 4.1** (Interpretation). We say that a statistic has an *interpretation* if it formalizes an intuitive concept and if the relationship between the statistic and the formalized intuitive concept is clear.

## 4.1. Formalizing intuitive concepts

How can we determine whether a given statistic has an interpretation? It turns out that the relationship between intuitive concepts and their formalizations has long been studied in philosophy, where this kind of problem is referred to as conceptual analysis (Chalmers, 1996; Chalmers & Jackson, 2001; Jackson, 1998; Justus, 2012). A central insight in this literature is that the intuitive concept is informal and pre-theoretic, whereas the formalized concept is theoretic and part of a workable theory (Carnap, 1945; 1950). Since

the intuitive concept is not formalized, there is no strict way that we could prove that a statistic formalizes an intuitive concept. However, there are arguments for whether a formalization captures an intuitive concept that come close to proving such statements.

Two important strategies to argue that a statistic formalizes an intuitive concept are the *convergence argument* and the *argument by derivation*.[3] Under the convergence argument, one proves that different attempted formalizations $X_1, X_2, X_3$ of an intuitive concept $x$ are equivalent. A famous example of a convergence argument is the formalization of the intuitive concept of computability, where different attempted formalizations (Turing computable, $\lambda$-expressible, general recursive function) were shown to be equivalent (Turing, 1937; Church, 1936; Gödel, 1931). Under the argument by derivation, one starts by arguing that any formalization of the intuitive concept should have properties A, B, and C, and then shows that there is a single formalism that satisfies all these desirable properties. A famous example of an argument by derivation is Cox's theorem used to argue for probabilism: that conditional Kolmogorov probabilities adequately capture the intuitive notion of degrees of belief given evidence (Cox, 1946; Paris, 1994; Williamson, 2010).

The main point here is not to provide a detailed step-by-step derivation for how the interpretations of traditional statistical quantities are obtained. This would be outside the scope of this work. Instead, we want to convince the reader that the question whether a statistic has an interpretation is a *fundamental* and, indeed, very *relevant* question. This is because only statistics with an interpretation can be reliably employed in real-world contexts.

## 4.2. Statistics of functions that have an interpretation

There are many examples of statistics of functions that have an interpretation. If the generalization error is low, a novel observation will probably be correctly labeled. If the degree of adversarial robustness is small, we can modify the input in subtle ways that change the prediction of the classifier (Goodfellow et al., 2014). More generally, simple statistics of functions that map the entire function to a single number often have an interpretation. Further examples are fairness metrics (Barocas et al., 2023), precision, recall, and other performance metrics for classifiers.

---

[3]This is far from exhaustive. There are at least two more relevant arguments, namely the *argument by replacement* (known as 'argument by interpretation' (Williamson, 2010)), for example the Dutchbook argument for probabilism (Ramsey, 1926; De Finetti, 1937) and the *argument by the consequences* as used by Shannon (1948) as complementary for his formalization of Information by Entropy. More generally, Carnap's method of explication is one of the most famous investigation into conceptual analysis.

Examples of post-hoc explanation methods that have an interpretation are *counterfactual explanations* (Wachter et al., 2017) and *anchors* (Ribeiro et al., 2018). Counterfactual explanations formalize the intuitive concept "What would need to change to get a different prediction?". Anchors formalize the intuitive concept that "Similar observations should be treated similarly". What is important is that for both counterfactual explanations and anchors, it is clear which intuitive questions can and cannot be answered given the explanations. For example, a counterfactual explanation applies only to an individual data point and does not necessarily tell us how the input features of other points would need to be changed. Similarly, anchors only apply to a specific region of the input space.

In addition, the literature on the intersection of statistics and machine learning has introduced various notions of feature importance that clearly define what they mean by "important" (Strobl et al., 2008; Zhang & Janson, 2020; Verdinelli & Wasserman, 2024). This includes model reliance (Donnelly et al., 2023), conditional model reliance (Fisher et al., 2019), and a variety of other notions (Lei et al., 2018). For an introduction, see Ewald et al. (2024). While the correct interpretation of the respective feature importance scores can sometimes be quite technical (compare Section 6.4), the papers in this literature take the matter of interpretation seriously. As such, they present positive examples of statistics of functions that have an interpretation. Interestingly, these notions of feature importance usually concern the *global* importance of the variable (Covert et al., 2020).

### 4.3. Many post-hoc explanations lack an interpretation

While there are examples of post-hoc explanations that have an interpretation (Section 4.2), unfortunately, many of them do not. According to Definition 4.1, this means that either (a) there is no intuitive concept that the mathematical formalism of the explanation relates to or (b) the relationship between the formalism and the intuitive concept is not sufficiently clear.

As an example, consider local feature attribution methods as they are being discussed in the literature on explainable machine learning (Ribeiro et al., 2016; Lundberg & Lee, 2017; Selvaraju et al., 2017). These methods address the question, "Which features are important for the prediction?". As such, there is an intuitive concept that the different methods attempt to formalize. However, the problem is that the relationship between formalism and intuitive concept is not sufficiently clear. This is because there are many different feature attribution methods, and different methods provide very different explanations (Bordt et al., 2022; Krishna et al., 2024). Consequently, it simply cannot be the case that all the different methods formalize the same intuitive concept. Applying the terminology introduced in

Section 4.1, we observe that feature attribution methods do not exhibit convergence, but *divergence*.

Unfortunately, the lack of interpretation does not only apply to a few methods and not only to feature attributions. Take almost any recent paper in the machine learning literature that introduces a new explainability method or claims to improve an existing one. The abstract and introduction will usually start by noting how important explainability is. The paper itself will then be about some technical derivations and experiments. In the end, the paper claims to have contributed to explainability. However, the evidence for this is usually lacking. This is because the paper never discussed the matter of interpretation: Does the proposed method formalize an intuitive concept? Can the method contribute to answering any specific questions that a human decision-maker might have?

## 5. Errors in Interpretation

We have argued that post-hoc explanations are statistics of functions, many of which lack an interpretation (Section 4.3). In this section, we argue that this is a significant problem. This is because explanation algorithms without interpretation lead to errors in interpretation.

### 5.1. Lessons from applied statistics

An important lesson in applied statistics is that the correct interpretation, even of relatively simple statistics, can be surprisingly difficult to achieve and that errors in interpretation occur frequently (Reid, 1954; Romeijn, 2022; Wasserstein & Lazar, 2016). This is especially the case when statistics such as confidence intervals or $p$-values are being interpreted by scientists from other disciplines or even by decision makers who are not scientists.

For an example of a common error in interpretation, consider the $p$-value as it arises in frequentist hypothesis testing. The $p$-value is a dataset statistic (Definition 2.2) that can be interpreted as the probability of observing an outcome at least as extreme as the observed dataset, assuming that the null hypothesis is true (Wasserstein & Lazar, 2016). This valid interpretation of the $p$-value requires imagining repeated experiments, even if only a single experiment has been conducted. Even though this valid interpretation exists, the $p$-value is often endowed with other invalid interpretations (Goodman, 2008; Colquhoun, 2014; Wasserstein & Lazar, 2016), especially Bayesian ones (Rice, 2018). For example, the $p$-value is sometimes wrongly interpreted as the probability that the null hypothesis is true (Romeijn, 2022, Sec. 3.2).

The literature on applied statistics knows many other examples of invalid interpretations of popular statistics, such as assuming that a 95% confidence interval contains the model

parameter with a probability of 95% or that an experiment that does not reject the null hypothesis shows that the null hypothesis is true (both of which are wrong) (Wasserstein & Lazar, 2016).

### 5.2. Statistics without interpretation lead to errors in interpretation

Considering that even relatively simple statistics that have an interpretation are sometimes endowed with other invalid interpretations (Section 5.1), it is not surprising that many of the complex statistics that have been proposed in the field of explainable machine learning have frequently been misinterpreted. What makes matters worse is that many post-hoc explanation algorithms lack an interpretation in the first place (Section 4.3).

Many papers discuss failure modes of explanation algorithms (Adebayo et al., 2018; Kumar et al., 2020; Molnar et al., 2020; Bordt et al., 2022; Geirhos et al., 2024; Bove et al., 2024; Huang & Marques-Silva, 2024; Tomaszewska & Biecek, 2024). This literature is full of examples of invalid interpretations, for example

1. Assuming that zero feature attribution means that a feature is irrelevant for the prediction (Bilodeau et al., 2024), or that zero feature attribution to a protected variable implies that a classifier is fair (Slack et al., 2020).

2. Assuming that explanation algorithms can reliably detect outliers or identify spurious correlations (Adebayo et al., 2020; 2022).

3. Assuming that explanations reveal causal relations in the real world (Molnar et al., 2020).

While many papers show how explanations should not be used and interpreted, it is rather hard to find papers that clearly show how explanation algorithms *should* be used (see, however, Murdoch et al. (2019)).

**We believe that the lack of an interpretation for the output of many popular algorithms is one of the main problems in the current literature on explainable machine learning**. This is because statistics without interpretation lead to errors in interpretation, especially when they are being used by scientists from other disciplines who might be less aware of their technical limitations.

## 6. What Explainable Machine Learning can Learn from Applied Statistics

In the previous sections, we have argued that explanation algorithms are statistics of functions, just like traditional statistics are statistics of probability distributions and datasets

(Section 2). We have also argued that the key question to ask about any given statistic is whether it admits an interpretation, that is, does it have a clearly specified relationship to an intuitive concept (Section 4). Taken together, our position is that explainable machine learning is an attempt to apply statistics to (learned) functions (Section 3). Consequently, explainable machine learning as a field has much to learn by considering research practices in applied statistics. In this section, we highlight some of the most important points.

### 6.1. Instead of asking whether a model is "interpretable" or whether a given statistic is an "explanation", we should ask which specific questions can be answered given a model or statistic

Useful statistics allow us to answer well-motivated and well-specified questions, but a single statistic will never allow us to answer *all* possible questions (Molnar et al., 2020; Freiesleben & König, 2023; Biecek & Samek, 2024). This is true in applied statistics, and it is also true in explainable machine learning.

Let us for a change illustrate this point with interpretable models, specifically the interpretable model class of Generalized Additive Models (GAMs) (Lou et al., 2013). GAMs represent the learned function as a set of univariate functions, where each function depicts the additive effect of one feature in the model. Given a GAM, certain specific questions can be directly answered by inspecting the model, whereas other questions cannot. Illustrative examples of questions that can be answered are "Are patients with pneumonia rated as high-risk?" and "How would the prediction change in response to a change in the input features?" (Caruana et al., 2015; Bosschieter et al., 2024). On the other hand, "Would the model discriminate against the members of a historically disadvantaged group in deployment?" is an example of a question that can *not* be answered by inspecting the interpretable model. In order to address this question, one would need to employ additional statistics, for example fairness metrics (Barocas et al., 2023).[4]

The key insight regarding future research in explainable machine learning is that papers should make explicit which *specific* questions a method is designed to answer (of course, this is directly related to our argument in Section 5).

### 6.2. Multifaceted intuitive concepts like trust are unlikely to be formalized by explanations

In Section 4, we have discussed that research often starts with intuitive concepts and attempts to formalize them. This raises the question: Which intuitive concepts can be suc-

---

[4]One of the advantages of interpretable model over post-hoc methods is that it is often fairly clear which specific questions can be answered given an interpretable model. See also Rudin (2019).

cessfully formalized? Of course, providing a general answer to this question is beyond the scope of our work. However, what stands out is that many of the intuitive concepts that motivate research in explainable machine learning are incredibly broad (consider, for example, the famous example of "trust" from the LIME paper (Ribeiro et al., 2016)). We would argue that a comparison with applied statistics clarifies that explainable machine learning is unlikely to succeed at formalizing concepts as broad as "trust" or "understanding" in individual statistics.

### 6.3. Explanations cannot replace other statistics such as fairness and robustness

Fairness and robustness measures are statistics of functions (Definition 2.3). This means that they are fundamentally similar to explanations, even if often perceived differently. Moreover, fairness and robustness measures usually have an interpretation (Section 2).

What does our analysis imply for the relationship between these measures and explanations? Most importantly, it implies that it is *not* the purpose of explanations to determine fairness (even though this is sometimes suggested in the literature, see, e.g. Slack et al. (2020)). This is because explanations should be designed to address specific questions (Section 6.1), and we already have statistics that formalize fairness notions. More generally, if we are interested in a particular property of a classifier and already have a statistic that formalizes it, then we should rely on this statistic instead of attempting to derive it implicitly through general-purpose explanations. This point is especially relevant for model auditing and regulation (Bordt et al., 2022; Cherian & Candès, 2024).

However, explanations might be helpful to debug classifiers to better understand why they are unfair (Pradhan et al., 2022).

### 6.4. Working with explanations requires expertise

In the statistical sciences, it is well-recognized that working with statistics requires expertise (Tishkovskaya & Lancaster, 2012). For example, we educate students to understand experimental design and correctly interpret $p$-values and the $F$-statistic. Because explanations are at least as complex as these traditional statistics, we believe that they will primarily be helpful to experts. Concretely, many currently popular methods, such as SHAP and gradient-based explanations, have complex technical properties and limitations related to technical notions such as in- and out-of distribution (Anders et al., 2020; Taufiq et al., 2023). Working with these explanations thus requires data science training. This does not necessarily have to be the case for all explanations — counterfactual explanations, in particular, can be interpreted without any specialized knowledge. Neverthe-

less, the analogy with applied statistics demonstrates that working with explanations, like other statistical quantities, requires expertise.

### 6.5. Benchmark datasets can help to probe explanations' empirical properties but generally do not provide interpretations

Recently, many have proposed benchmarks to evaluate explanations (Hooker et al., 2019; Pawelczyk et al., 2021; Hedström et al., 2023). Arguably, benchmarks can be a helpful tool to probe the empirical properties of explanations, perhaps somewhat similar to simulation studies in statistics.[5] Most importantly, standardized benchmarks improve upon the research paradigm where papers introducing novel algorithms also introduce novel evaluation metrics.

At the same time, it is important to recognize that benchmarks are unlikely to provide interpretations for the output of post-hoc explanation algorithms. This is because interpretation is about the relationship between statistics and intuitive human concepts (see Section 4.1). To make this more concrete, let us again rely on the analogy between explainable machine learning and applied statistics and ask how interpretations of traditional statistics are obtained. At the risk of oversimplification, let us discuss the mean and the median. It is well-known that both the mean and the median are useful for interpreting probability distributions (for example, in the analysis of income distributions (Saez & Zucman, 2016)). At the same time, the limitations of these statistics are also clearly understood. For example, the mean is not robust to outliers, and neither statistic captures the skewness of the data. Now, the idea of collecting some representative probability distributions to benchmark the interpretative performance of the mean against the median subject to some externally defined metric seems quite absurd. The same is true for other statistics of probability distributions. By analogy, we argue this is also true for statistics of functions.

---

[5] One might even argue that many explanation algorithms are so complex and intransparent that they are themselves black boxes, and that the literature has implicitly realized this by studying them as such. Consider, for example, the literature that aims to make sense of the empirical properties of saliency maps (Adebayo et al., 2018; Hooker et al., 2019; Adebayo et al., 2020). In this literature, a saliency map is simply a black box algorithm that delivers an input attribution map over an image. What is more, not only the literature on saliency maps fall into the category, but almost every paper that empirically measures whether explanation algorithms can perform some task ("detect spurious correlations", "highlight the most important feature") implicitly engages in the approach of studying explanation algorithms as black boxes (Adebayo et al., 2022).

## 7. What about Mechanistic Interpretability?

To be concise, this paper has focused chiefly on post-hoc explanation algorithms. However, we believe that the analogy between explainable machine learning and applied statistics extends to the entire field. As another example, let us briefly outline how it applies to the increasingly popular field of mechanistic interpretability (Saphra & Wiegreffe, 2024; Sharkey et al., 2025).

Mechanistic interpretability aims to reverse-engineer the inner workings of neural networks (Elhage et al., 2021). It analyzes patterns in model weights and activations (Nanda et al., 2023) using a variety of methods, for example, probes (Gurnee et al., 2023), or by training sparse auto-encoders (Lieberum et al., 2024; Karvonen et al., 2024).

What is the relationship between mechanistic interpretability and the arguments outlined in this paper? We would argue that the relationship between applied statistics and mechanistic interpretability is even more apparent than for post-hoc explanations. This is because many papers in mechanistic interpretability literally apply traditional statistical methods to model weights and activations. Consider, for example, Nanda et al. (2023), where a key technique is to show that the Fourier decomposition of model weight matrices is sparse.

More generally, the weights and activations of a neural network fall under the concept of a statistic of a function (Definition 2.3). The same holds for derived statistics such as dimensionality reduction techniques and probes. However, note that there is an interesting difference between post-hoc explanation methods and mechanistic interpretability: Whereas post-hoc explanations offer a fixed statistic to be applied to all functions, researchers in mechanistic interpretability often aim to *find* a statistic that can provide insights about a given function. Nevertheless, once a statistic is chosen, the questions that should be asked about the statistic are the same as for post-hoc explanations (Section 4): Does the statistic formalize any intuitive concepts? What concrete questions can be answered with the statistic? In some cases, for example, when a probe relates to an interpretable outcome (Karvonen et al., 2024), mechanistic interpretability has found good answers to these questions.

## 8. Related Work

There is a large diversity of viewpoints regarding explainable machine learning, and much debate about its possible use cases (Bansal et al., 2021; Ghassemi et al., 2021; Bordt et al., 2022). Various authors have offered conceptualizations of the field, for example, by incorporating insights from the social sciences (Doshi-Velez & Kim, 2017; Selbst & Barocas, 2018; Gilpin et al., 2018; Miller, 2019; Leavitt & Morcos, 2020; Freiesleben & König, 2023; Wang et al.,

2023). Lipton (2018) offers an early and well-known critique of the field. Murdoch et al. (2019) propose a systematic framework to evaluate and select explanations.

Recently, researchers have begun to take a more statistical approach towards explainable machine learning. For example, Bilodeau et al. (2024) study feature attribution methods via their ability to reject hypotheses about model behavior. Sharma et al. (2024) connect practices in explainable machine learning with p-hacking (Stefan & Schönbrodt, 2023), and Senetaire et al. (2023) formulate feature importance as a statistical inference problem. Cherian & Candès (2024) study fairness auditing from a statistical perspective, and Scholbeck et al. (2024) connect practices in explainable machine learning with sensitivity analysis. To the best of our knowledge, this position paper is the first to make the case that explainable machine learning is applied statistics for learned functions.

An interesting connection exists between our work and the literature on human-centered explainability (Liao et al., 2020; Liao & Varshney, 2021; Ehsan et al., 2021). While we are skeptical that algorithmic explanations can serve as "an essential requirement for people to trust and adopt AI deployed in numerous domains" (Liao & Varshney, 2021), we argue that the literature should focus on the real-world questions that explanations are designed to answer (Section 4). Interestingly, this claim is somewhat similar to the idea that the explanation should focus on the needs of the human users, a tenet in the literature on human-centered explainability (Liao & Varshney, 2021). For example, Figure 1 in Liao et al. (2020) is conceptually similar to the list of intuitive human questions that we provide in Section 4.

## 9. Alternative Views

The most important alternative view to our position is the idea that algorithmic explanations of machine learning systems should be similar to the explanations that humans provide about their own behaviour.[6] According to this perspective, algorithmic explanations would allow humans to understand the behaviour of a machine learning model in the same way that we can understand a human's behaviour by asking them to explain it (for example, by stating the factors relevant to their decision). While it is rarely explicitly expressed in research papers (see, however, Phillips et al. (2021, Section 8)), this alternative view has had a significant influence on the field of explainable machine learning. In particular, it underlies the perspective that explainable

---

[6]By attempting to predict machine behavior through postulated intentions, this view takes an "intentional stance" towards machine learning systems. While this approach might be useful to understand human behavior, it is questionable whether it is helpful for algorithms. See, however, Dennett (1987) for a defense of the intentional stance towards early algorithms.

machine learning can attain the same broad-based goals as human explanations, including successful collaboration with humans (Kim, 2015; Lakkaraju et al., 2022), trust (Ribeiro et al., 2016), or general understanding of model behaviour (Hagras, 2018). We suspect that the choice of the terminology "explanation" in the literature on explainable machine learning contributes to this perspective because it naturally invokes similarities with other forms of explanations, such as human or scientific explanations (Woodward & Ross, 2022).

By contrast, we argue that the analogy between algorithmic and human explanations is misguided, that explanations are better understood as statistics, and that future research in explainable machine learning should look for inspiration in statistics research rather than elsewhere. We also argue that explanations are complex technical tools that might mostly be useful to experts.

## 10. Discussion

In this work, we propose a novel perspective on explainable machine learning. By contrast to many other perspectives on the field, we focus less on what explainable machine learning *could be* and more on what actually *has been* done in the literature over the last decade. Our central insight is that the field attempts to develop useful statistics of high-dimensional decision surfaces and that this is remarkably similar to "traditional" statistical approaches. Consequently, we argue that explainable machine learning should be seen as applied statistics for learned functions.

**Where do we go from here?** In our view, a significant amount of research in explainable machine learning over the last decade has been confused by overambitious goals that the relatively simple methods that have been proposed in the literature could simply not deliver. While our paper is critical of these practices, the purpose of our paper is not to discourage future research. On the contrary, we believe that the field of explainable machine learning studies important research questions that are worth investigating. By advancing the analogy between explainable machine learning and applied statistics, we hope to set more realistic expectations for what the field can and cannot deliver. In particular, **the analogy between explainable machine learning and applied statistics can be useful when communicating about explainable machine learning with researchers from other disciplines**. This is because many have at least a high-level understanding of what it means to conduct applied statistical research.

## Impact Statement

Explainable machine learning is often seen as one of the key components of responsible machine learning. The purpose of our position paper is to encourage discussion about this important topic. As such, we don't believe that our paper will have negative societal consequences.

## Acknowledgements

We would like to thank Julius Adebayo, Timo Freiesleben, Gunnar König, Suraj Srinivas, Bob Williamson and anonymous reviewers for helpful comments and discussion. This work has been supported by the German Research Foundation through the Cluster of Excellence "Machine Learning - New Perspectives for Science" (EXC 2064/1 number 390727645), the CZS Institute for Artificial Intelligence and Law, and the WIN program of the Heidelberg Academy of Sciences and Humanities, financed by the Ministry of Science, Research and the Arts of the State of Baden-Württemberg.

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
