# OpenReview forum: "Position: Rethinking Explainable Machine Learning as Applied Statistics"
_ICML.cc/2025/Position_Paper_Track — ICML 2025 Position Paper Track poster_

### Official Review · Reviewer_Ci2w · 2025-03-07

**Significance:** 3
**Argument Clarity:** 2
**Rating:** 3
**Confidence:** 3

**Questions:**

- How do the authors reconcile the logical inconsistency between convergence, intuitiveness, and requiring expertise to handle the complexity of explanations?
- Can you please justify why your notion of "formalizing intuitive concepts" is not also superficial?
- How does an applied researcher answer the question in section 6.1?
- If section 6.2's claim is true, how do we distinguish between whether an intuitive concept is too broad or not?

**Discussion Potential:**

3

**Ethical Review Concerns:**

I do not believe there are any ethical concerns with this paper.

**Paper Summary:**

This paper argues the following:
1. Explanations are statistics of a given model-distribution-instance combinations. Explanations map high-dimensional objects to low-dimensional mappings
2. However, a model-summary being a statistic alone is insufficient for claiming it is an explanation. The summary should also "formally" connect to an "intuitive" concept
3. Formalizing the definition of an explanation resolves many debates within the field

The paper tries to argue these three points by examples to the current literature.

**Position:**

Yes

**Position In Title:**

Yes

**Related Work:**

2

**Strengths And Weaknesses:**

## Strengths
- Claim 1 is well-justified as it uses a rigorous argument connecting the rigorous definition of a statistic to explanations.
- Raising the question, "what specific questions can be answered given a model or statistic?" is very interesting and well-argued. The example with GAMs is very clear and highlights the strengths and limitations of interpretations of GAMs. The XAI community would benefit from answering such questions in their papers to mitigate the problems introduced in Section 5.

## Weaknesses
I do not feel that claims 2 and 3 are not thoroughly justified, and I hope the authors can clarify my concerns about their argument.

### Issues with Claim 2
    - The authors first acknowledge that working with explanations requires expertise because "explanations are complex technical tools" (line 412). They also later claim that feature attribution methods are not explanations because multiple competing feature attribution methods arrive at differing conclusions of the intuitive concept, "which features are important for the prediction?" (lines 225-226). This argument of feature attribution methods simplify the complexity of feature attribution methods by not highlighting that the different feature attribution methods have different notions of "important." We could apply this same logic to counterfactual explanations, which the authors laud as being "explanations [that] formalize the intuitive concept 'What would need to change to get a different prediction?'" (lines 202-203). However, if we changed the underlying distance metric used in learning counterfactuals (e.g., change from using Euclidean distance to (1 - cos similarity) ), then we could get potentially differing counterfactuals. According to the authors feature attribution argument, a valid explanation must be (1) consistent with other explanations aiming to answer an "intuitive" concept; and (2) the "intuitive" concept should not require "any specialized knowledge" (line 346). However, all explanations require specialized knowledge for interpretation, so the authors logic seems to suggest that _no_ current explanation in the literature is actually an explanation. This logical inconsistency stems from my second issue with the argument (next bullet).
    - The authors criticize human-centered explanations for being "superficial" (line 408). But because there is no rigorous, logical, or axiomatized definition for an explanation "formalizing intuitive concepts," this philosophical condition seems entirely superficial as well. It would be most helpful to the XAI community if the authors could offer a set of axiomitized or rigorous sufficient conditions for an explanation being an explanation. The authors are on the right direction by using formal definitions of a statistic, but the authors need more on the philosophical lens.

### Issues with Claim 3
- The authors provide an alternative definition of an explanation to the already many definitions that exist in the literature (the introduction of Lipton's mythos of interpretability paper introduces 5 differing notions already). The authors do not attempt to reconcile existing debates; instead, they denigrate existing perspectives by claiming they are "superficial."


### Post-rebuttal review
Thank you for engaging in the discussion. I now understand what literature you were targeting by "feature attribution methods" -- this makes sense. As you have committed to clarifying these issues, I believe my main concerns have been addressed. As such, I am happy to raise my score.

Other notes
I do not see how some papers support your argument, e.g. Xue et al (2023). This is an attribution-agnostic framework, so I do not see how this paper connects to a bad definition of "importance." Please go through the references you just shared and make sure that the papers fit into the discussion.

**Support:**

2

---

> ### Author Rebuttal · Authors · 2025-03-31
>
> Thank you for the detailed review and insightful questions. Below, we provide a detailed rebuttal to your questions/comments.
>
> *“Formalizing the definition of an explanation resolves many debates within the field”*
>
> This is not what we say in the paper. To clarify, we argue that recognizing the parallels of explainable machine learning and applied statistics (L 14-16) would resolve debates (L. 084-085).
>
> Claim 2: *“different feature attribution methods have different notions of "important."*
>
> This is an excellent counter-argument. We have ourselves believed for quite some time in this resolution. However, we now see the following problem: We have no clarity of the different notions of “importance” of the various methods.
>
> If we could spell out which variants of “importance” are formalized by LIME, SHAP, etc., then we could say that, after all, they formalize a kind of cloud of the question “what features are important?” However, precisely what “importance” means has never been successfully specified for any of these methods. And as long as this has not been specified, the argument in Section 4.3 of our paper holds.
>
> To further clarify, our point is not that it would be impossible to come up with more precise concepts of “feature importance” and to develop methods that formalize such concepts. In fact, this approach is exactly what we argue for in Section 6.1!
>
> Claim 2: *“We could apply this same logic to counterfactual explanations“*
>
> Another interesting counter-argument. Let us outline why we hold that all counterfactual explanations have an interpretation, unlike feature attribution methods. The reason is that no matter the distance metric and other computational details (of which we agree there are many), the prediction will change when we change the input as specified in the explanation. That is no matter how a counterfactual explanation is computed, it always has the simple, valid interpretation that when we change the input, the prediction changes. In the explainability literature, this aspect seems somewhat unique to counterfactual explanations. The same cannot be said about feature attribution methods.
>
> Claim 3:   *“The authors criticize human-centered explanations for being "superficial" (line 408) … do not attempt to reconcile existing debates; instead, they denigrate existing perspectives by claiming they are "superficial."”*
>
> We apologize for this misunderstanding. We would never claim that human-centered explanations or other ideas to conceptualize explainable machine learning are superficial. Indeed, we think that other perspectives on explainable machine learning hold great value, and we specifically believe that focusing on the explanatory needs of the human who receives the explanation is worthwhile (see L. 413- 417 in our paper).
>
> We thank the reviewer for spotting this glitch and commit to rephrase L 407-411.
>
> *“How do the authors reconcile the logical inconsistency between convergence, intuitiveness, and requiring expertise to handle the complexity of explanations?”*
>
> We don’t think that there is a logical inconsistency. Concretely, we would argue that
>
> - There are intuitive questions that can be understood with little or no expertise (for example, “Which input features do I need to change to change the prediction?’’)
> - Other intuitive questions require a certain degree of expertise to understand the involved concepts (for example, ``Is the model relying on spurious artifacts in the input image?’’)
> - Some explanations can be understood with little or no expertise (for example, counterfactual explanations).
> - Many other explanations are more complex, and working with them requires expertise (Section 6.4)
> - Arguing that an explanation formalizes an intuitive concept, for example, using a convergence argument, always requires expertise.
>
> *“How does an applied researcher answer the question in section 6.1?”*
>
> Section 6.1 sets a new research agenda. We ask the XAI community to present their explanation algorithms together with concrete questions that the algorithms are supposed to answer. We argue that this is necessary to move towards a more rigorous evaluation of the merit of explanation algorithms, focused on whether they allow us to answer specific questions or not.
>
> *“If section 6.2's claim is true, how do we distinguish between whether an intuitive concept is too broad or not?”*
>
> As we see it, providing a good answer to this question is the core research agenda of explainable machine learning. In other words, the only way to answer this question is to attempt to develop algorithms that formalize different concepts on a case-by-case basis (and to learn when such attempts tend to fail). As we write in Section 6.2, we believe that many of the concepts that are thrown around in the literature are too broad for any single algorithm to adequate capture. However, this hypothesis of ours could be falsified by newly proposed algorithms.
>
> We would be happy to answer any additional questions.

---

> > ### Comment · Reviewer_Ci2w · 2025-04-02
> >
> > I thank the authors for the interesting responses and the valuable discussion.
> >
> > - If we could spell out which variants of “importance” are formalized by LIME, SHAP, etc., then we could say that, after all, they formalize a kind of cloud of the question “what features are important?” However, precisely what “importance” means has never been successfully specified for any of these methods. And as long as this has not been specified, the argument in Section 4.3 of our paper holds.
> >   - This is a very interesting point, but I think generalizing LIME/SHAP's arguable lack of interpretation to all feature attribution methods is a very strong and, frankly, false claim. I could agree with LIME's claim, and SHAP technically does have an interpretation but it is too complex for most practitioners to remember. But there are many methods in the "etc" component that do rigorously define interpretations. For example, model reliance and conditional model reliance have all very clearly defined "importance:" model reliance measures a feature's importance by how much model performance drops when the feature is perturbed, while conditional model reliance accounts for feature dependencies to isolate a feature's unique contribution and has been widely used across statistics and machine learning [1, 2]. Moreover, multiple sections of these papers introducing new feature attribution methods focus on the interpretation of the statistics [3, 4]. And each of these methods clearly define what they mean by "important." These papers are, overall, in support of your argument that xAI should acknowledge their connections to statistics as these papers are all written by statisticians. I recommend the authors tone down the universality of statements and instead focus on specific examples of feature attribution methods for this section.
> > - Another interesting counter-argument. Let us outline why we hold that all counterfactual explanations have an interpretation, unlike feature attribution methods. The reason is that no matter the distance metric and other computational details (of which we agree there are many), the prediction will change when we change the input as specified in the explanation. That is no matter how a counterfactual explanation is computed, it always has the simple, valid interpretation that when we change the input, the prediction changes. In the explainability literature, this aspect seems somewhat unique to counterfactual explanations. The same cannot be said about feature attribution methods.
> >  - Again, the same could not be said about e.g. LIME and SHAP. But certainly the same could be said about permutation importance or conditional model reliance; even if you change specific hyperparameters (e.g., the choice of loss function), the overall interpretation still holds. E.g., model reliance measures a feature's importance by how much model performance drops when the feature is perturbed no matter how you define "performance." I think this discussion is worth adding to the paper -- I believe these nuances could yield very interesting discussions to the xAI community.
> >
> > - I have a remaining question: The authors' claim about _all_ feature attribution methods seems misguided, as I hopefully elucidated. Do you have more concrete examples of existing explanations that are not explanations that you could rigorously argue?
> >
> > ## References
> >
> > 1. Fisher, Aaron, Cynthia Rudin, and Francesca Dominici. "All models are wrong, but many are useful: Learning a variable's importance by studying an entire class of prediction models simultaneously." Journal of Machine Learning Research 20.177 (2019): 1-81.
> > 2. Donnelly, Jon, et al. "The rashomon importance distribution: Getting rid of unstable, single model-based variable importance." Advances in Neural Information Processing Systems 36 (2023): 6267-6279.
> > 3. Verdinelli, Isabella, and Larry Wasserman. "Decorrelated variable importance." Journal of Machine Learning Research 25.7 (2024): 1-27.
> > 4. Zhang, Lu, and Lucas Janson. "Floodgate: inference for model-free variable importance." arXiv preprint arXiv:2007.01283 (2020).

---

> > > ### Author Response · Authors · 2025-04-05
> > >
> > > Thank you for engaging in the discussion!
> > >
> > > *“I think generalizing LIME/SHAP's arguable lack of interpretation to all feature attribution methods is a very strong and, frankly, false claim. [...] For example, model reliance and conditional model reliance have all very clearly defined "importance" ”*
> > >
> > > You are right to point out that there exist feature importance measures that clearly define importance.
> > >
> > > Let us perhaps begin by clarifying the literature on “explainable machine learning” that we are targeting with our paper. As we see it, this literature started with the deep learning revolution and papers such as Simonyan et al. (2013), Zeiler and Fergus (2014), and Bach et al. (2015). It gained a lot of prominence around 2016-2018 when LIME, SHAP, and Integrated Gradients were published (among others). Ever since then, this literature has been quite active (e.g. Xue et al. (2023), Deiseroth et al. (2024)). It is this literature that we have in mind with our claim that “feature attribution methods lack an interpretation”.
> > >
> > > Now, the reviewer points out that statisticians have also been concerned with feature importance and that their papers clearly specify what they mean by the concept. We completely acknowledge that the reviewer is right on this point. It is also true that these methods often fall within the formal (Section 2) and conceptual (Section 4) framework that we develop in our paper, which means that we agree that a discussion of them should be added.
> > >
> > > As suggested by the reviewer, we commit to clarify that our claim about feature attribution methods in Section 4.3 applies specifically to certain methods that have been developed in the literature on explainable machine learning (Ribeiro et al., 2016; Lundberg & Lee, 2017; Selvaraju et al., 2017, Chattopadhay et al., 2018). In addition, we commit to discuss positive examples of feature attribution methods that have an interpretation, including the ones suggested by the reviewer, in Section 4.2. We agree with the reviewer that this is a rather interesting discussion (also concerning your point on model reliance and counterfactual explanations), and we commit to reflect this discussion in the paper.
> > >
> > > *“These papers are, overall, in support of your argument that xAI should acknowledge their connections to statistics as these papers are all written by statisticians.”*
> > >
> > > Indeed, this is the point that we want to make!
> > >
> > > *“Do you have more concrete examples of existing explanations that are not explanations that you could rigorously argue?”*
> > >
> > > For feature attribution methods, we would point the reviewer to (Selvaraju et al., 2017, Chattopadhay et al., 2018, Tsang et al. 2020, Hesse et al. 2021, Xue et al. 2023.).
> > >
> > > For examples of other types of explanations, we would point the reviewer to “concepts learned by neural networks” (e.g. Ghorbani et al. 2019) and the debate whether attention coefficients in transformer models are “explanations” (e.g. Serrano and Smith (2019)).
> > >
> > > Concerning your question, note that there is an important subtlety: It is very difficult to argue that a given method does not have an interpretation. Therefore, the argument that we make in our paper is that for many popular methods in explainable machine learning, it has never been successfully argued that they have an interpretation and that this is a problem.
> > >
> > > Thank you again for your helpful review. We hope that we have addressed your most important concerns. If this is the case, we would kindly ask you to consider increasing your score.
> > >
> > >
> > > $\text{}$
> > >
> > >
> > > **Additional References:**
> > >
> > > (other references are in the paper)
> > >
> > > Simonyan et al.. "Deep inside convolutional networks: Visualising image classification models and saliency maps.," CoRR, abs/1312.6034, 2013.
> > >
> > > Zeiler and Fergus. "Visualizing and understanding convolutional networks." In 13th European Conference on Computer Vision, 2014.
> > >
> > > Sofia Serrano and Noah A. Smith. 2019. "Is Attention Interpretable?" ACL 2019
> > >
> > > Ghorbani et al., "Towards Automatic Concept-based Explanations" NeurIPS 2019
> > >
> > > Tsang et al., "How does This Interaction Affect Me? Interpretable Attribution for Feature Interactions", NeurIPS 2020
> > >
> > > Hesse et al., "Fast Axiomatic Attribution for Neural Networks", NeurIPS 2021
> > >
> > > Xue et al., "Stability Guarantees for Feature Attributions with Multiplicative Smoothing", NeurIPS 2023
> > >
> > > Deiseroth et al, "ATMAN: Understanding Transformer Predictions Through Memory Efficient Attention Manipulation", NeurIPS 2024

---

### Official Review · Reviewer_3kCK · 2025-03-09

**Significance:** 3
**Argument Clarity:** 4
**Rating:** 4
**Confidence:** 4

**Questions:**

See my previous comments.

**Discussion Potential:**

3

**Paper Summary:**

This paper argues that the field of explainable machine learning should be viewed through the lens of applied statistics. Particularly, applied statistics studies statistics of probability distributions and datasets (Definitions 2.1 and 2.2), and importantly engage in two distinct activities: first, study mathematical properties of these statistics, and second, provide arguments that they describe intuitive concepts in particular scientific settings. In a similar manner, explainable machine learning studies statistics of high dimensional functions (Definition 2.3), and consequently the two activities of applied statistics have parallels for explainable machine learning. However, the authors note that many post-hoc explanations fail the second task -- lacking no clear mapping between the formalism and an intuitive concept. The authors conclude by discussing several lessons for the explainable machine learning community that can be drawn from applied statistics.

**Position:**

Yes

**Position In Title:**

Yes

**Related Work:**

4

**Strengths And Weaknesses:**

Altogether, I found the position paper to be extremely well written and argued. It articulates a potentially foundational perspective for the field of explainable machine learning with clear arguments for a path forward.

I see three main weaknesses of / points to clarify in the position paper:

1) One of the key ideas in the position paper is that statistics have an ``interpretation'' if they formalize an intuitive concept and the relationship between an intuitive concept and the statistic is clear (Definition 4.1). The authors provide two examples of this definition, but for one of them ("convergence argument"), they do not even provide an example from statistics but instead from computability. This definition would be far more effective if the authors could provide further specific examples -- in other words, even as someone who does applied statistics for a living, I am not sure that I would be able to follow advice at this level of abstraction! It seems like the leading example the authors have in mind is a p-value (indeed, it is the leading example of an interpretation gone astray). But are there others in applied statistics?

2) There is an argument that the authors do not go far enough in their analogy between explainable machine learning and applied statistics.  In particular, after all, there is arguably no field of ``applied statistics'' -- instead, successful applied statistics is embedded within specific fields such as biostatistics, econometrics, quantitative political science, etc. Many of the successes of applied statistics come from formalizing intuitive concepts in specific fields. In economics, there are constant concerns about selection and endogeneity, and the field now uses causal inference frameworks to formalize these statements.

From this perspective, you could interpret the authors' position as: just like there is no separate field of applied statistics, there should be no separate field of explainable machine learning. Instead, the intuitive concepts that explainable machine learning must formalize is inherently domain/problem specific --- what constitutes an answer to the question "should I trust the predictions of the model" or "what input features are important to the prediction" is inherently domain/problem specific. The answers should look different for use in (i) hiring algorithms, (ii) consumer lending algorithms, (iii) computer vision algorithms, etc.

In fact, I thought this would have been where the authors ended up going, but instead they seemed to focus on more "general statistical" concepts as the parallel to explainable machine learning. In which case, I'm not sure I understand the distinction between applied statistics and just statistics in how the position.

3) The parallel of explainable machine learning as reporting functionals of learned prediction functions is arguably known, although perhaps it is never clearly articulated in a review paper. To give some examples, it is commonly formalized that an interpretation or an explanation is a statistic of the prediction function [e.g., 1,2]. Consequently, I don't really view the main point in Section 2 as wholly novel. At a minimum, the authors should explicitly discuss that this formalization of what is an explanation exists (although again perhaps not stated as directly or with this emphasis).

[1] Fisher, Rudin and Dominici. "All Models are Wrong, but Many are Useful: Learning a Variable's Importance by Studying an Entire Class of Prediction Models Simultaneously." JMLR 2019.

[2] Marx, Park, Hasson, Wang, Ermon and Huan. "But Are You Sure? An Uncertainty-Aware Perspective on Explainable AI." AISTATS 2023.

## Post Rebuttal Comments
I thank the authors for their detailed reply to my comments. I am positive about the position paper -- I think it articulates a potentially foundational perspective for XAI, and it could spark valuable discussion. If anything, as I stated in the review, I think the authors can go much further in drawing the analogy between statistics and its counterparts embedded within specific fields such as biostatistics, econometrics, quantitative political science, etc.

**Support:**

4

---

> ### Author Rebuttal · Authors · 2025-03-31
>
> We thank the reviewer for the insightful review and highlighting that our paper **“outlines a potentially foundational perspective for the field of explainable machine learning with clear arguments for a path forward.”** Below, we give detailed answers to your questions/comments.
>
> *“One of the key ideas in the position paper is that statistics have an ``interpretation'' if they formalize an intuitive concept [...] This definition would be far more effective if the authors could provide further specific examples”*
>
> Let us give a more statistical example that illustrates the relationship between formalizations and intuitive concepts: The Bayesians vs. Frequentists debate.
>
> First, note that we have an intuitive concept of probability (we say things like: “x is probable”, or “x is more probable than y”). It turns out that the intuitive concept of probability needs to be disambiguated into two more specific concepts, namely subjective probability (rational degree of belief) and objective probability (real world property) [Carnap: https://doi.org/10.2307/2269162), Gillies: https://doi.org/10.4324/9780203132241]. Both subjective probability and objective probability are examples of intuitive concepts.
> Now, the relevant formalization of the intuitive concept of probability is Kolmogorov probability. Interestingly, this is a single formalization that captures both subjective probability and objective probability [Hájek, https://plato.stanford.edu/entries/probability-interpret/]. To this day, researchers argue about the interpretation of the formalism (Bayesians vs. Frequentists).
>
> The key point here is that the Bayesians vs. Frequentists debate is an example of a debate about how formalism relates to intuitive concepts.
>
> We will add this example to the paper.
>
> *“It seems like the leading example the authors have in mind is a p-value (indeed, it is the leading example of an interpretation gone astray). But are there others in applied statistics?”*
>
> There are many examples of statistics and (invalid) interpretations – we decided to focus on the p-value for illustrative purposes and space issues. Perhaps one of the most prominent misinterpretations in statistics is when a sample correlation is given an invalid causal interpretation. Another example of a common misinterpretation is that one assumes after observing the mean of a probability distribution that there exist observations from the distribution that take the mean value.
>
> *“... the authors do not go far enough in their analogy [...] : just like there is no separate field of applied statistics, there should be no separate field of explainable machine learning. Instead, the intuitive concepts that explainable machine learning must formalize is inherently domain/problem specific --- what constitutes an answer to the question […] "what input features are important to the prediction" is inherently domain/problem specific.”*
>
> We completely agree with this great comment. In fact, we would see it as a natural prolongation of our arguments.
>
> *“[...] I thought this would have been where the authors ended up going, but instead they seemed to focus on more "general statistical" concepts as the parallel to explainable machine learning. In which case, I'm not sure I understand the distinction between applied statistics and just statistics in how the position”*
>
> Based on your comments, you understand our position very well.
>
> We wrote this position paper precisely because we believe that explainable machine learning needs to “rethink” itself as a discipline similar to biostatistics, econometrics, quantitative political science, etc. In our paper, we attempt to provide arguments for why explainable machine learning falls into the same category as these other disciplines. We decided to use the umbrella term “applied statistics” because we believe that it helps to convey our main message.
>
> We further agree with the reviewer that the questions asked in the different disciplines are domain-specific and that each discipline needs to formalize the intuitive concepts relevant to its own domain. “Feature importance” for machine learning models is an example of a concept specific to explainable machine learning, as is “relying on spurious artifacts”. Section 6.1 in our paper can be understood as arguing that explainable machine learning needs to focus on concrete, domain-specific research questions.
>
> *“The parallel of explainable machine learning as reporting functionals of learned prediction functions is arguably known, although perhaps it is never clearly articulated”*
>
> Thanks for the additional references. This is true, and we commit to acknowledge it in our paper.
>
> Thank you again for your high-quality review. We would be happy to answer any additional questions.

---

### Official Review · Reviewer_Gn4h · 2025-03-13

**Significance:** 2
**Argument Clarity:** 2
**Rating:** 2
**Confidence:** 2

**Questions:**

N/A

**Discussion Potential:**

2

**Paper Summary:**

This paper argues to think machine learning explanation in the way of applied statistics. To be honest, after reading this paper, I got confused on the position and I do not understand how machine learning explanation can be think the the way of applied statistics specifically.  Please treat my review as an educated guess given I don't get the core idea of this work totally.

**Position:**

Yes

**Position In Title:**

No

**Related Work:**

2

**Strengths And Weaknesses:**

Strength:
1. Great imagination on thinking explainable machine learning in the lens of applied statistics.

Weaknesses:
1. I don't get the idea. While I can see how statistics are often used in explaining ML model behaviour, I really cannot make the analogical connection.

**Support:**

1

---

> ### Author Rebuttal · Authors · 2025-03-31
>
> We thank the reviewer for honestly stating that their review is “an educated guess” and for granting us with "great imagination" on one of our main points.
>
> *“[...] after reading this paper, I got confused on the position and I do not understand how machine learning explanation can be think the the way of applied statistics [...] “*
>
> Let us try to restate the gist of our idea in simple terms.
>
> The mathematical form of applying an explanation algorithm to a machine learning model is analogous to using a traditional statistic for a probability distribution: The explanation algorithm summarizes and simplifies a complex object (Section 2 and Section 3). However, for a low-dimensional summary of a function to have explanatory value, it cannot just be a random functional. We need to know what the summary means. To capture this aspect of the explanation, we draw on works in the philosophy of science and say that the statistic needs to formalize an intuitive concept (Section 4). We also argue that formalizing intuitive concepts is what traditional statistics for probability distributions have been doing all along (in other words, we are looking for answers to intuitive human questions; see the beginning of Section 4). This establishes the analogy between explainable machine learning and applied statistics.
>
> Based on the analogy, we discuss various aspects of the explainability literature (Section 4.3, Section 5) and how we believe research practices can be improved (Section 6).
>
> We would be happy to answer any additional questions.

---

### Official Review · Reviewer_rPbx · 2025-03-14

**Significance:** 2
**Argument Clarity:** 2
**Rating:** 3
**Confidence:** 3

**Questions:**

1. The paper briefly touches on mechanistic interpretability but doesn't delve deeply into how this field aligns with the statistical perspective you propose. Could you expand on how mechanistic interpretability fits into the broader framework of XAI as applied statistics, and what challenges might arise in bridging these two areas?

2. In your perspective, how should researchers balance the trade-off between model accuracy and interpretability when designing XAI algorithms? Does the statistical perspective suggest new ways to approach this balance that could improve both?

**Discussion Potential:**

2

**Paper Summary:**

The paper argues for a reevaluation of explainable machine learning (XAI) from the perspective of applied statistics. It posits that post-hoc explanation algorithms should be understood as statistics of high-dimensional functions. In this analogy, explainable machine learning is positioned as a method of applying statistical techniques to learned functions, similar to traditional applied statistics which summarizes probability distributions or datasets. The paper emphasizes the importance of not only developing explanation algorithms but ensuring that these explanations have clear interpretations that are meaningful to human users, addressing real-world, intuitive questions.

The core critique of current XAI approaches is that many of the explanation algorithms lack a proper interpretation, leading to potential misinterpretations. The authors stress that, like traditional statistics, explanation methods should aim to formalize intuitive concepts clearly and respond to specific questions rather than attempt to broadly address vague goals such as "trust" in a model. The paper draws lessons from applied statistics to propose concrete improvements in research practices, urging for a more focused, question-oriented approach in XAI, and recommends using benchmarks to evaluate empirical properties while being cautious about deriving interpretations directly from them.

**Position:**

Yes

**Position In Title:**

Yes

**Related Work:**

3

**Strengths And Weaknesses:**

## Strengths:
The paper introduces a fresh viewpoint by directly connecting explainable machine learning to applied statistics, offering a clear framework to improve the field.

## Weaknesses:
1. Regarding Interpretability, while the paper presents a strong argument for the application of applied statistics to XAI, it may not address all aspects of explainability, such as its psychological or social implications, which are also crucial for understanding human-AI collaboration. In fact, aligning the models' behaviour with the human value is crucial for current AI researches, which could be beyond the scope of applied statistics.

2. The paper assumes that working with XAI requires expert knowledge, which may limit the accessibility of these tools for non-experts or those in fields outside of data science or machine learning.

3.  Some readers might find the heavy reliance on mathematical formalism a barrier to understanding, particularly those new to applied statistics or machine learning.

**Support:**

2

---

> ### Author Rebuttal · Authors · 2025-03-31
>
> Thank you for reviewing our paper and appreciating our fresh viewpoint. Below, we provide a detailed rebuttal to your questions/comments.
>
> *“the paper [...] may not address all aspects of explainability, such as its psychological or social implications”*
>
> Indeed, we highlight explicitly that we do not focus on these aspects [L.66-72], as they have been extensively discussed in prior works [e.g., Miller, 2019] and lie somewhat outside the scope of our work. Of course, we agree with the reviewer that psychological and social aspects of explainability are relevant!
>
> *“aligning the models' behaviour with the human value is crucial for current AI research, which could be beyond the scope of applied statistics.”*
>
> This is precisely what we ask and argue about in our paper. “Is the AI aligned with human values?” is a perfect example of an intuitive question that humans have about machine learning models (compare the beginning of Section 4). The crucial point that we discuss in our paper is: How do we know that a given statistic of a machine learning model addresses such a question? (Compare Section 4.1. Of course, this is highly complex for the alignment problem). As we argue in Section 6.2, we are skeptical that (individual) statistics of machine learning models can be said to answer intuitive questions that are as broad-based as alignment with human values.
>
> *“The paper assumes that working with XAI requires expert knowledge, which may limit the accessibility of these tools [...]”*
>
> Yes, but this is not a limitation of our paper but of the discussed explainability methods! In our view, this limitation is not properly pointed out in the literature, so highlighting and discussing it is an important contribution of our paper.
>
> *“Could you expand on how mechanistic interpretability fits into the broader framework of XAI as applied statistics, and what challenges might arise in bridging these two areas?”*
>
> While the methods used in mechanistic interpretability are somewhat different from post-hoc XAI, they are also statistical tools and similar to traditional statistical methods. This means that we could as well write a variant of Section 2 highlighting the formal similarities between statistics and mechanistic interpretability instead of statistics and post-hoc methods. From this starting point, the remaining arguments would essentially be the same (what intuitive questions does a given technique in mechanistic interpretability answer?).
>
> A challenge that might arise in bridging the two areas is that the objects studied in mechanistic interpretability (the activation patterns of neural networks) might have different empirical properties than the objects studied in other areas of applied statistics (say large-scale datasets). This means that despite the general usefulness of the analogy, mechanistic interpretability still “needs to solve its own problems” (note that this holds true for XAI in general).
>
> *“[...] how should researchers balance the trade-off between model accuracy and interpretability when designing XAI algorithms? Does the statistical perspective suggest new ways to approach this balance[...] ?”*
>
> Great question. To quote Section 6.1 in our paper: “Instead of asking whether a model is “interpretable” [...] we should ask what specific questions can be answered given a model”.
>
> This means that when we build an interpretable model, we should not focus on its "interpretability" as an abstract concept or an inherent property of the model. Instead, we should ask: What do we want to do with the model? Are there any particular questions that we would like to answer? Does the model that we are building allow for this kind of analysis? (consider the example of GAMs discussed in Section 6.1).
>
> Depending on the application, there might be trade-offs with accuracy (or not; we believe this is an empirical question). The key point, however, is that we should not focus on the abstract concept of an accuracy-interpretability trade-off but on a concrete exercise of model-building that obtains certain accuracy levels and allows us to answer certain kinds of questions.
>
> For an example of a recent paper that illustrates this approach quite well, see
>
> Ismail et al., Concept Bottleneck Language Models For protein design, ICLR, 2025 (https://arxiv.org/abs/2411.06090)
>
> We would be happy to answer any additional questions.

---

> > ### Comment · Reviewer_rPbx · 2025-04-08
> >
> > Thanks for your rebuttal. I would like to point out several literatures on balancing the performance and efficiency of attribution-based methods like Shapley values. Please add these papers in the related works. Currently, I would like to raise my score to 3 for supporting this paper.
> >
> >
> > [1] Jethani N, Sudarshan M, Covert I C, et al. Fastshap: Real-time shapley value estimation[C]//International conference on learning representations. 2021.
> > [2] Covert I, Kim C, Lee S I. Learning to estimate shapley values with vision transformers[J]. arXiv preprint arXiv:2206.05282, 2022.
> > [3] Wang S, Tang H, Wang M, et al. Gnothi Seauton: Empowering Faithful Self-Interpretability in Black-Box Models[J]. arXiv preprint arXiv:2410.21815, 2024.

---

> > > ### Author Response · Authors · 2025-04-08
> > >
> > > We thank the reviewer for providing the references and for increasing their score. We commit to include the references in the paper.

---

### Decision · Program_Chairs · 2025-04-30

**Decision:**

Accept (poster)

**Comment:**

The point that the authors make is very interesting, and the statistical methods that are used to study natural processes can be clearly used to study ML models. The paper should lead to useful discussions, and hopefully bring ML and stats communities closer.